# Inhibition Mechanism of *Lactiplantibacillus plantarum* on the Growth and Biogenic Amine Production in *Morganella morganii*

**DOI:** 10.3390/foods12193625

**Published:** 2023-09-29

**Authors:** Zhenxiao Sun, Yi Zhang, Xinping Lin, Sufang Zhang, Yingxi Chen, Chaofan Ji

**Affiliations:** 1School of Food Science and Technology, Dalian Polytechnic University, Dalian 116034, China; 211720860000985@xy.dlpu.edu.cn (Z.S.);; 2State Key Laboratory of Marine Food Processing and Safety Control, Dalian Polytechnic University, Dalian 116034, China

**Keywords:** fermented foods, *Lactiplantibacillus plantarum*, *Morganella morganii*, inhibit, biogenic amines, RNA-seq

## Abstract

*Morganella morganii*, a spoilage bacterium in fermented foods, produces harmful biogenic amines (BAs). Although *Lactiplantibacillus plantarum* is widely used to inhibit spoilage bacteria, the inhibition pattern and inhibition mechanism of *M. morganii* by *Lpb. plantarum* are not well studied. In this study, we analysed the effects of the addition of *Lpb. plantarum* cell-free supernatant (CFS) on the growth and BA accumulation of *M. morganii* and revealed the mechanisms of changes in different BAs by using RNA sequencing transcriptome analysis. The results showed that *Lpb. plantarum* CFS could significantly inhibit *M. morganii* BAs in a weak acid environment (pH 6), and the main changes were related to metabolism. Carbohydrate and energy metabolism were significantly down-regulated, indicating that *Lpb. plantarum* CFS inhibited the growth activity and decreased the BA content of *M. morganii*. In addition, the change in histamine content is also related to the metabolism of its precursor amino acids, the change in putrescine content may also be related to the decrease in precursor amino acid synthesis and amino acid transporter, and the decrease in cadaverine content may also be related to the decrease in the cadaverine transporter. The results of this study help to inhibit the accumulation of harmful metabolites in fermented foods.

## 1. Introduction

Biogenic amines (BAs) are small molecule compounds that are formed in protein-rich food, particularly fermented foods. BAs are formed from free amino acids (FAAs) by microbial decarboxylation [1]. The excessive intake of BAs is harmful to human health and can cause food poisoning with symptoms such as nausea, diarrhoea, and headache [2]. The presence of other amines has been shown to increase histamine toxicity [3]. As consumers become more concerned about food safety, the use of safe and environmentally friendly biological methods to suppress BA levels in food is gaining acceptance [4].

BAs, especially histamine, are associated with some of the spoilage microorganisms and can be used as quality indicators of the freshness and safety of fermented food products [5]. A variety of BAs producing microbial populations are present in food products, such as *Morganella morganii*, *Klebsiella pneumoniae*, *Escherichia* spp., and *Pseudomonas* spp. [6]. Among these species, *M. morganii* is a typical substance BA-producing strain that produces large amounts of histamine, which is considered an indicator of food spoilage [7]. There are many factors that influence the production of BAs by *M. morganii* in food. It has been shown that *M. morganii* is sensitive to acidic environments, and its ability to produce BAs is reduced in low acid environments that are unfavourable for its growth [8].

*Lactiplantibacillus plantarum* (formerly *Lactobacillus plantarum*) is a lactic acid bacterial species used as the starter for a variety of fermented food [9]. *Lpb. Plantarum* produces a variety of antimicrobial substances. Metabolites of *Lpb. Plantarum*, such as organic acids, fatty acids, bacteriocins, and other bacteriostatic compounds [10], have demonstrated the capability to inhibit spoilage bacteria [11]. Additionally, the cell-free supernatant (CFS) of lactic acid bacteria is easily accessible, eliminating the need for complex extraction processes. Consequently, this CFS has garnered attention from researchers in the field of food safety due to its antibacterial properties [12]. Until now, the inhibitory capacity of *Lpb. Plantarum* has been mainly focused on common spoilage organisms [13], such as *Enterobacter cloacae*, *Pseudomonas luteola*, and *Photobacterium damselea*, whereas few studies on the inhibition of *M. morganii* have been reported. We previously isolated a strain of *Lpb. plantarum* from fermented fish, which demonstrated an inhibitory capacity against *M. morganii* [14]. It does not directly degrade BAs, but rather inhibits the growth of *M. morganii*. However, unlike other spoilage organisms, studies of the inhibitory mechanisms against *M. morganii* have not elucidated. Thus, the pattern and mechanism of the effect of *Lpb. plantarum* on the growth and BA metabolism of *M. morganii* need to be further investigated.

In this study, we first investigated the effects of *Lpb. plantarum* M1 cell-free supernatant and acids on growth and BA accumulation in *M. morganii* YC-16. Then, the transcriptome sequencing of the expressed genes (DEGs) in *M. morganii* samples was conducted. Then, subsequent bioinformatics-based analyses revealed the inhibitory mechanism of *Lpb. plantarum* cell-free supernatant on *M. morganii* at the gene transcription level. The aim of this study is to provide new insights into the inhibition mechanism of BAs by *Lpb. plantarum* and to provide a theoretical basis for improving the safety of fermented food products.

## 2. Materials and Methods

### 2.1. Strain and Growth Conditions

The strain *Lpb. plantarum* M1 and *M. morganii* YC-16 (School of Food Science and Technology, Dalian Polytechnic University, Dalian, China) were screened from fermented food and conserved at −80 °C. *Lpb. plantarum* M1 was grown statically in de Man, Rogosa, Sharp (MRS) broth medium (Qingdao Hope Bio-Technology Co., Ltd., Qingdao, China) at 37 °C for 24 h to obtain preculture. *M. morganii* YC-16 was grown statically in tryptic soy broth (TSB) medium (Qingdao Hope Bio-Technology Co., Ltd., Qingdao, China) at 30 °C for 24 h to obtain preculture. The strains were continuously activated and cultured for two generations.

### 2.2. Preparation of Lpb. plantarum Cell-Free Supernatant (CFS)

The activated *Lpb. plantarum* M1 (OD600 approximately 1.6) was inoculated into MRS broth medium (Qingdao Hope Bio-technology Co., Ltd., Qingdao, China) at 2% (*v*/*v*) inoculum for 16 h at 37 °C and then centrifuged at 6200× *g* for 10 min (Sorvall ST8, ThermoFisher Scientific Co., Ltd., Shanghai, China) at 4 °C to collect the bacteria. The collected bacteria were added to 1/10 of the original volume of fresh MRS broth (At this point, the inoculum density is approximately 10^10^ CFU/mL) at 37 °C for 16 h, and then centrifuged at 6200× *g* for 10 min at 4 °C. The supernatant was filtered through a 0.22 μm sterile aqueous filter membrane to remove other debris to obtain a CFS.

### 2.3. Effect of Different Conditions on the Growth and BA Production of M. morganii

The CFS of *Lpb. plantarum* M1 was mixed with TSB medium (Qingdao Hope Bio-technology Co., Ltd., Qingdao, China) at a ratio of 1:1 (*v*/*v*) and pH adjusted to 4, 5, 6, and 7, filtered through a sterile 0.22 μm aqueous filter membrane and prepared for use. The activated *M. morganii* was inoculated with 2% (*v*/*v*) of the medium, and 300 μL of the mixture was added to the culture well plate at 30 °C for 48 h. The growth dynamics of *M. morganii* was measured using a fully automated growth curve analyser (Bioscreen C, Oy Growth Curves Ab Ltd., Turku, Finland).

Immediately after the growth curve determination, the culture broth was collected and centrifuged at 9600× *g* for 5 min (Sorvall ST8, Thermo Fisher Scientific Co., Ltd., Shanghai, China) in a high-speed refrigerated centrifuge, and the supernatant was retained. The samples were pretreated with reference to the BA determination method of Zhang [15], and the BA content was determined by using ultra high-performance liquid chromatography (UPLC; Nexera LC-30A system, Shimadzu, Kyoto, Japan) combined with a triple quadrupole mass spectrometer (Qtrap 5500, AB Sciex, Toronto, ON, Canada).

### 2.4. Transcriptome Sequencing Sample Preparation

*M. morganii* YC-16 cultured in TSB medium at pH 7 was used as a control group (CON). In the first experimental group (PON), the pH of TSB medium was adjusted to 6, and *M. morganii* YC-16 was inoculated with 2% (*v*/*v*) inoculum for two generations and incubated at 30 °C. The second experimental group (PCFS) was inoculated with CFS of *Lpb. plantarum* M1 mixed with TSB medium in a ratio of 1:1 (*v*/*v*) and adjusted to pH 6. *M. morganii* YC-16 was inoculated after two generations of activation at an inoculum size of 2% (*v*/*v*) at 30 °C. After incubation for approximately 10 h, 500 μL of the bacteria were transferred to sterile and enzyme-free centrifuge tubes and centrifuged at 10,000× *g* for 10 min (CF16RN, Hitachi, Ibaraki-ken, Japan) at 4 °C. The supernatant was discarded, and the bacteria were snap-frozen in liquid nitrogen and stored at −80 °C in an ultra-low temperature refrigerator for subsequent transcriptome sequencing.

### 2.5. RNA Extraction, cDNA Synthesis, Library Construction and Sequencing

RNA extraction and high-throughput RNA sequencing (RNA-seq) were performed by Shanghai Ouyi Biomedical Technology Co. (Shanghai, China) Total RNA was extracted using a TIANGEN RNAprep Pure Bacterial Total RNA Extraction Kit (Tiangen, Beijing, China). Bacterial samples (1 × 10^8^ CFU) were thoroughly resuspended with a 100 μL Tris-EDTA buffer containing lysozyme, centrifuged with 350 μL lysate shaking, and then the supernatant was taken. The supernatant was mixed well with 250 μL anhydrous ethanol and transferred to RNase-Free column CR3 (adsorbent column was placed in a collection tube) and the supernatant was discarded after centrifugation (13,400× *g*, 1 min), and then 350 μL deproteinizing solution was added and the supernatant was discarded after centrifugation (13,400× *g*, 1 min). A total of 80 μL of DNase working solution (10 μL of DNase I plus 70 μL of DNA Digest buffer) was added to the centre of the adsorbent column CR3, before it was left at room temperature (RT) for 15 min. A total of 350 μL of deproteinization solution was added, then it was centrifuged and the supernatant was discarded. A total of 500 μL of rinse solution was added to the adsorbent column CR3, where it was left at RT for 2 min, before it was centrifuged and the supernatant was discarded. A total of 500 μL of rinse solution was added again, it was left at RT for 2 min, centrifuged for 2 min, the waste solution was poured off, and the adsorbent column CR3 was left at RT for several minutes to dry the residual rinse solution thoroughly. The adsorption column CR3 was transferred into a new RNase-Free centrifuge tube, 50 μL RNase-Free ddH_2_O was added dropwise to the middle part of the adsorption membrane, it was left at RT for 2 min, and then centrifuged at 12,000 rpm for 2 min to obtain the RNA solution.

The ribosomal RNA was digested with Illumina’s Ribo-Zero Magnetic Kit (Bacteria, MRZB12424) and broken into short fragments by adding interruption reagents. The interrupted RNA is used as a template to synthesize a one-stranded cDNA using a six-base random primer, and then a two-stranded cDNA is synthesized by preparing a two-stranded synthesis reaction system. dUTP is used instead of dTTP in the synthesis of the cDNA second strand, and then the strand containing dUTP is digested using the UNG enzyme method, and only the cDNA strand with different connectors is retained. The cDNA strand is purified using the kit, and the purified cDNA strand is then end-repaired, A-tailed, and sequenced, the fragment size is selected, and PCR amplification is performed. The constructed RNA library was quality checked with an Agilent 2100 Bioanalyzer and sequenced using the Illumina NovaSeq platform.

### 2.6. RNA-Seq Data Analysis

Using Trimmomatic [16] software (version 0.35, http://www.usadellab.org/cms/index.php?page=trimmomatic (accessed on 3 September 2023)), the raw data generated in high-throughput sequencing with spliced, low-quality reads were filtered to obtain high-quality clean reads (valid data). The clean reads obtained for CON, PON, and PCFS were sequenced against a specified reference genome using Rockhooper [17] (version 2.0.3, http://cs.wellesley.edu/~btjaden/Rockhopper (accessed on 3 September 2023)) to obtain information on the position on the reference genome or gene, as well as information on the sequence features specific to the sequenced sample. In calculating the expression differences of genes, the number of reads matched to genes in each sample was obtained using Rockhooper [17] software, and the raw data were normalized by the RPKM [18] method (reads per kilobase per million mapped reads). RPKM represents the number of reads per kilobase per million reads from a gene by dividing the number of reads matched to a gene by the number of reads matched to the genome (in millions) and the length of the RNA (in KB). The number of counts for each sample gene was normalised using DESeq (version 1.8.3, http://www-huber.embl.de/users/anders/DESeq (accessed on 3 September 2023)) [19] software (basemean values were used to estimate expression), the differential FoldChange values were calculated, and the number of reads was tested for significance of differences using NB (negative binomial distribution test). DEGs were selected according to log2FoldChange > 1 or log2FoldChange < −1 with *p* < 0.05. Meanwhile, gene ontology (GO) (http://geneontology.org (accessed on 3 September 2023)) enrichment analysis of differentially expressed genes was performed and pathway analysis was performed using the Kyoto Encyclopedia of Genes and Genomes (KEGG) (http://www.kegg.jp/kegg (accessed on 3 September 2023)) [20] database to determine the biological functions or metabolic pathways that differentially genes mainly affect.

### 2.7. Statistical Analysis

All analyses were carried out in triplicates. The arithmetic mean and standard error of the mean was calculated using the MS Office Excel 2021 package. Data were analysed by using one-way ANOVA when conditions of homogeneity of variance were present. *p*-values < 0.05 were considered to be statistically significant.

## 3. Results and Discussion

### 3.1. Effect of Lpb. plantarum CFS on M. morganii Growth and BA Production

*M. morganii* is sensitive to acidic environments, and overly acidic environments are unfavourable for its growth and affect the production of BAs [21]. When the pH was decreased from 7 to 4 (Figure 1A), the growth of *M. morganii* was inhibited, leading to a gradual decrease in the OD600 from 1.45 to 0.03. The comparison of Figure 1A,B reveals that the growth of *M. morganii*, when cultured under various conditions, gradually transitions from the logarithmic growth phase to the stable phase in approximately 10 h. Furthermore, it is evident that the inhibitory impact of the *Lpb. plantarum* metabolites on the growth of *M. morganii* is more pronounced under identical pH conditions. When *Lpb. plantarum* CFS was added to the *M. morganii* (Figure 1B), the growth of *M. morganii* decreased by 19.15% at pH 7, whereas *Lpb. plantarum* CFS inhibited *M. morganii* by more than 50% at both pH 6 and pH 5 conditions. It was found that the inhibitory substances in the *Lpb. plantarum* CFS remained active in an acidic environment and only partially inhibitory after pH neutralisation [22]. This result suggests that there are bacteriostatic agents in the *Lpb. plantarum* CFS used in this study that can effectively inhibit the growth of *M. morganii*.

Subsequently, we determined the changes in the content of BAs generated by *M. morganii* under different culture conditions. In the control group (as shown in Figure 1 and Appendix A), the content of BAs produced by *M. morganii* was significantly reduced under pH 5 and pH 4 conditions compared to the pH 7 culture condition, but *M. morganii* produced more BAs at pH 6. This may be due to the inhibition of the growth of BA-producing strains in the weak acid environment, which induces the secretion of the amino acid decarboxylase to decarboxylate free amino acids and produce alkaline biogenic amines to resist the influence of the acidic environment on the growth of the strains [23]. In the groups supplemented with *Lpb. plantarum* metabolites (Figure 1D and Appendix A), the content of the BAs exhibited a decline as pH decreased. Moreover, the total BA content was lower than that of the control group at equivalent pH conditions. At pH 6, *Lpb. plantarum* CFS displayed the most notable inhibitory impact on the BA production capability of *M. morganii*, resulting in a substantial 44.66% reduction in total BAs. Notably, the greatest reduction was observed in cadaverine content, reaching a decrease of 74.60%. Simultaneously, the production of histamine witnessed a decrease of only 24.90%. This suggests that the inhibitory patterns of *Lpb. plantarum* CFS on the metabolism of various types of biogenic amines in *M. morganii* differ, and further research is required to explore this issue.

### 3.2. Overview of the Transcriptional Analysis and Differentially Expressed Genes

Transcriptome analyses of *M. morganii* were performed using the Illumina NovaSeq platform under three conditions: blank group (CON), acidic environment (PON), and the combination of *Lpb. Plantarum* CFS with acidic environment (PCFS). Each RNA-seq library yielded an average of 8 million reads. The alignment rate of the nine samples with the reference library ranged from 73% to 80%, indicating satisfactory sequencing results. Principal component analysis (PCA) plots (Figure 2A) revealed notable inter-sample variability while demonstrating high reproducibility among the same sample set in three parallel runs. This substantiated the reliability of the RNA-seq data.

Pairwise comparisons of transcriptome sequencing data were performed to identify genes exhibiting significant expression differences. The criteria for differential gene selection were set at a *p*-value < 0.05 and an absolute value of |log2 FoldChange| > 1, with a higher FoldChange value signifying greater expression disparity between the two samples.

In the PCFS vs. CON group, a total of 1218 differentially expressed genes were identified (464 with up-regulated expression and 754 with down-regulated expression). Similarly, in the PON vs. CON group, 1300 differentially expressed genes were found (612 with up-regulated expression and 688 with down-regulated expression). For the PCFS vs. PON group, a total of 1305 differentially expressed genes (568 up-regulated and 737 down-regulated) were identified. The overall differences in expression levels among the three groups are illustrated in the volcano plot (Figure 2).

### 3.3. Functional Analysis and Classification of DEGs

To acquire a thorough comprehension of the functional categories associated with *M. morganii* DEGs under various culture conditions, we conducted an analysis of the GO enrichment. This analysis was structured around three primary categories: biological processes, cellular components, and molecular functions. As shown in Figure 3, the effects of culture environments on BA production by *M. morganii* exhibited differences. In the comparison between the PCFS group and the control group (Figure 3A), the most significant differences were found in the tricarboxylic acid cycle, chemotaxis, arginine biosynthesis process, glycolytic process and quinone process. The most significant difference between the PON group and the control group was evident within the structural component of the ribosome (Figure 3B). Subsequently, differences were also observed in translation, arginine biosynthetic processes and ribosomal and trans-membrane transporter activities. The tricarboxylic acid cycle, translation, ribosome, structural component of the ribosome and quinone binding showed the most significant differences between the PCFS group and the PON group (Figure 3C).

The expression pathways of DEGs were subsequently elucidated using KEGG biopathway classification and enrichment analysis. KEGG pathway classification maps were generated through paired comparisons of samples from CON, PCFS, and PON groups (Figure 4). In the KEGG enrichment categories, the metabolism category displayed the highest expression across the three experimental conditions, comprising 91.34% (PCFS vs. CON), 87.03% (PCFS vs. PON), and 88.27% (PON vs. CON) of the total expression in each respective comparison. In addition, DEGs are also associated with membrane transport, signal transduction, and other functional categories. These encompass pathways like ABC transporters, two-component systems, and a range of other pathways.

### 3.4. Analysis of DEGs Associated with M. morganii BA Production Inhibition

#### 3.4.1. Carbohydrate Metabolic Pathways

Carbohydrate metabolism encompasses all the biochemical processes responsible for the formation, breakdown, and interconversion of sugars to ensure a constant supply of energy to living cells [24]. Glycolysis (ko00010) and the tricarboxylic acid (TCA) cycle (ko00020) pathways were significantly affected when *M. morganii* was cultured in an environment containing *Lpb. plantarum* CFS and acidic conditions. Glycolysis is the conversion of glucose to pyruvate through a series of enzymatic reactions and the release of energy. Compared to the CON group, the most important rate-limiting enzyme *pfk* (fructose phosphokinase) during glycolysis was significantly up-regulated by 9.6-fold in the PCFS group. Most other genes were also significantly up-regulated (Figure 5A). In the subsequent oxidation of pyruvate, the expression of *adh* (ethanol dehydrogenase) experienced a remarkable 159.4-fold up-regulation, making it the gene with the highest up-regulation in carbon metabolism in *M. morganii* cells. The enhancement of these processes stimulates the glycolytic pathway, potentially serving as a protective or resistant mechanism for *M. morganii* to safeguard its cells against potential harm from *Lpb. plantarum* CFS. The TCA cycle is a central metabolic pathway that is not only the main source of energy for the organism, but also a hub for converting three major substances: sugars, lipids, and proteins [25]. In the TCA cycle, the expression levels of almost all genes (Figure 5B), including *gltA*, *icd*, *sucA*, and *sucB*, responsible for encoding the key rate-limiting enzymes of the citrate synthase, isocitrate dehydrogenase, and alpha-ketoglutarate dehydrogenase, displayed a decline subsequent to treatment with *Lpb. plantarum* CFS. This observation suggests a down-regulation of the TCA cycle pathway in *M. morganii* treatment with *Lpb. plantarum* CFS.

The 2-methylcitrate cycle (MCC), an interconnected cycle with close ties to the TCA cycle, is widely found in bacteria and plays a role in the breakdown of propionic acid or propionyl-CoA (ko00620). Bacteria with the MCC have the capability to utilise odd fatty acids, cholesterol, and other substrates capable of producing propionyl-CoA as a carbon source [26]. When the cycle is blocked, the bacteria experience an inability to grow and reproduce. In this study (Figure 5C), the expression levels of the three key enzymes *prpB* (2-methylcitrate Synthase), *prpC* (2-methylcitrate Dehydratase) and *prpD* (isocitrate methyl ester lyase) were significantly down-regulated after *Lpb. plantarum* CFS treatment, resulting in a blockage of the entire MCC. Impaired MCC function often leads to overaccumulation of the toxic metabolite 2-methylcitric acid, which in turn inhibits the activity of fructose-1,6-bisphosphate kinase, a key enzyme in the gluconeogenic pathway [27]. This leads directly to the bacteria growing stunted. This is consistent with the results of this study where the expression of *fbp* (fructose-1, 6-bisphosphatase), a gene related to the glycolytic pathway, was down-regulated.

However, unlike the PCFS group, the PON group showed enhanced overall carbohydrate metabolism and up-regulation of genes in the glycolytic and TCA cycle pathways in *M. morganii* compared to the CON group. These up-regulations of genes in the glycolysis and TCA cycle promote increased metabolic activity to generate more energy and prevent the acidic environment from causing damage to the bacterial cells. The diminished growth activity in *M. morganii* could potentially be attributed to an overall attenuation of the TCA cycle. The differential genes associated with the TCA cycle were significantly down-regulated in the PCFS group compared to the PON group. Therefore, we speculate that this is mainly due to the presence of antimicrobial peptides and other inhibitory substances in *Lpb. Plantarum* CFS [14] that inhibit carbohydrate metabolism and thus the growth of *M. morganii*.

#### 3.4.2. Energy Metabolic Pathways

Changes in the growth environment can seriously affect the energy balance of the cells, requiring additional energy to maintain or restore it. Similar to carbohydrate metabolism, *Lpb. plantarum* CFS and acidic environment culture can affect *M. morganii*’s energy metabolism. Oxidative phosphorylation (ko00190) is an intracellular biochemical process that generates proton gradients through the oxidation of substances. This process ultimately results in the conversion of ADP to ATP. Compared to the CON group (Figure 6A), the PCFS and PON groups showed an up-regulation of *ndh*, a gene related to NADH oxidative metabolism, which facilitated the conversion of NADH to NAD^+^ [28]. Succinate quinone oxidoreductase is central to cellular metabolism and energy conversion, contributing to the tricarboxylic acid cycle and the electron transport chain. It catalyses the oxidation of succinate to fumarate. The gene *sdhA*, a regulator of succinate quinone oxidoreductase, was down-regulated in the PCFS group compared to the CON group, while there was no change in the PON group. Thus, the energy pathway of *M. morganii* was weakened by the acidic environment and *Lpb. plantarum* CFS treatment, and the expression of DEGs were significantly higher in the PCFS group than in the PON group. In the acidic environment, the rate of tyramine reduction after the addition of *Lpb. plantarum* CFS in cultures was approximately the same as the rate of growth inhibition by *M. morganii*. We speculate that the decrease in this BA accumulation may be mainly related to the inhibition of *M. morganii* cell growth by the addition of *Lpb. plantarum* CFS.

#### 3.4.3. Amino Acid Metabolic Pathways

BAs are biologically active organic compounds formed from the decarboxylation of precursor amino acids by the action of amino acid decarboxylases produced by microorganisms [29]. As a result, the levels of precursor amino acids are highly likely to influence the production of biogenic amines by microorganisms. The different BAs are formed from distinct amino acids through the action of their corresponding amino acid decarboxylases. Tyramine, tryptamine, phenylethylamine, histamine, and cadaverine are the products of decarboxylation catalysed by tyrosine decarboxylase, tryptophan decarboxylase, phenylalanine decarboxylase, histidine decarboxylase, and lysine decarboxylase respectively. Conversely, the synthesis of putrescine, spermidine, and spermine involves a comparatively intricate, multi-step process [30]. In *M. morganii* cells treated with *Lpb. plantarum* CFS and an acidic environment, the biosynthesis and metabolic pathways of several amino acids were significantly down-regulated in amino acid metabolism (Figure 7). This is likely to have an impact on *M. morganii* ’s capacity to generate BAs. The histidine biosynthesis pathway (ko00340) is affected by *hisG* (ATP phosphoribosyltransferase), *hisH* (imidazole glycerol-phosphate synthase subuni), *hisC* (histidinol-phosphate aminotransferase), and *hisD* (histidinol dehydrogenase). Compared to the PON group (Figure 7F), *hisG*, *hisH*, *hisC*, and *hisD* were up-regulated in the PCFS group. Histidine is an essential precursor for the synthesis of histamine and this up-regulation most likely led to an increase in the levels of histamine precursor amino acids. *hdc* (histidine decarboxylase) is a key enzyme in the decarboxylation of histidine for the synthesis of histamine. *hdc* was up-regulated 4.6-fold in the PCFS group compared to the PON group. *hdc* up-regulation promotes the decarboxylation of histidine to produce histamine. The addition of *Lpb. plantarum* CFS to the culture of *M. morganii* in an acidic environment resulted in a decrease in histamine levels, but the decrease in histamine levels was significantly less than the inhibition of cell growth of *M. morganii*. Therefore, changes in the histamine levels of *M. morganii* cultured with the addition of *Lpb. plantarum* CFS correlated with both cell growth activity and amino acid metabolism in this experiment. In tyrosine metabolism (ko00350), tryptamine metabolism (ko00380), phenylalanine metabolism (ko00360), and phenylalanine, tyrosine, and tryptophan biosynthesis (ko00400), the major DEGs expression in both experimental groups (PON and PCFS) was associated with precursor amino acid synthesis of bioamines compared with the CON group. The down-regulation of *trpB* (tryptophan synthase beta chain) results in reduced tryptophan levels (Figure 7A). The decreased expression of *pheA2* (prephenolic acid dehydratase) and *hisC* contributes to lower phenylalanine levels, while the down-regulation of *hisC* also leads to diminished tyrosine levels (Figure 7B). In *M. morganii* subjected to acidic environment treatment, seven genes within the arginine synthesis pathway (ko00220) exhibited substantial down-regulation. Specifically, the expression of *argA* (amino acid N-acetyltransferase), *argB* (acetylglutamate kinase), argH (argininosuccinate lyase), and *argD* (acetylornithine aminotransferase) were down-regulated during the process of converting glutamate to ornithine (Figure 7C). In the conversion of ornithine to arginine pathways, the expression of *argF* (ornithine carbamoyltransferase), *argG* (argininosuccinate synthase), and *argH* (argininosuccinate lyase) displayed down-regulation (Figure 7E). The decreased expression of these genes indicates a constriction in the pathway for ornithine and arginine synthesis. Compared with the CON group, the down-regulation amplitude of differentially expressed genes in the arginine synthesis pathway in the PCFS group was two-fold greater than that in the PON group. The up-regulation of *glnA* (glutamine synthetase) led to a reduction in the utilization of glutamate for ornithine conversion (Figure 7D). Therefore, *M. morganii* treated with *Lpb. plantarum* CFS showed more pronounced inhibitory effects on the production of putputamide than before treatment, which may be related to the inhibition of the amino acid synthesis of its precursor.

In the lysine synthesis pathway of *M. morganii* (ko00310), the production of lysine was inhibited by the down-regulation of *lysA* (diaminopyrimidine decarboxylase) in the PON group compared to that in the CON group. The up-regulation of *lysA* in the PCFS group promoted lysine production compared with the PON group (Figure 7G). The accumulation of cadaverine in *M. morganii* was drastically reduced by *Lpb. plantarum* CFS treatment compared to culture under acidic conditions only, and the rate of cadaverine decrease was significantly higher than the growth inhibition rate of *M. morganii*. However, despite this, lysine production did not exhibit a direct positive correlation with cadaverine accumulation. Thus, we speculate that changes in cadaverine accumulation might be linked to the cadaverine exporter, *CadB* [31].

#### 3.4.4. Membrane Transporters-Related Pathways

ABC family transporters (ko02010) are membrane-integrated proteins that catalyse the hydrolysis of ATP and use the energy generated to facilitate the transmembrane transport of substrates, including sugars, amino acids, and proteins, and play an important role in life activities [32]. Changes in the monosaccharide transport proteins affected carbohydrate metabolism (Figure 6B). Compared with the CON group, the genes associated with the monosaccharide transport proteins, *RbsB*, *LsrA*, *LsrB*, *LsrC*, *LsrD*, and *XltC*, were being down-regulated in the PCFS group, but up-regulated in the PON group. The phosphate transporter protein genes *PstA* and *PstB* were up-regulated in the CFS group to promote Pi synthesis and transport to the cytoplasmic phosphate transport system [33]. Phosphorus uptake and utilisation plays an important role in biological processes, such as organismal genetics, energy metabolism, cell membrane integrity, and intracellular signal transduction [34]. The differential genes related to monosaccharide transporters and phosphate transporters complement the above changes in carbohydrate metabolism and energy metabolism pathways, and can more fully express the bacteriostatic substances in *Lpb. plantarum* CFS, which has an important impact on the BA production ability of *M. morganii*.

The membrane transport systems on the cell surface also have an Impact on the levels of precursor amino acids for biogenic amines within the cells. Compared to the control group, the expression of glutamate transporter protein-related genes (*GltI*, *GltK*, *GltJ*, *GltL*) were down-regulated in the PCFS and PON groups, and the expression of relevant DEGs in the PCFS group was higher than that in the PON group. Therefore, the decrease in putrescine may also be related to the down-regulation of amino acid transporter-related genes. Transcriptomic data showed that genes involved in membrane transport in *M. morganii* were significantly down-regulated after *Lpb. plantarum* CFS treatment and the reduction of amino acid transport proteins limited BA production.

## 4. Conclusions

Lactic acid bacteria cell-free supernatant (CFS) possesses the ability to inhibit the growth of foodborne spoilage bacteria. This study investigated the antibacterial properties of *Lpb. plantarum* CFS against *M. morganii*, a bacterium known to produce BAs in fermented foods. Additionally, a systematic analysis of the inhibitory patterns of CFS on the growth of *M. morganii* and various BA metabolisms was conducted using transcriptomics methods. In the future, experiments should be conducted to investigate the inhibition of BA metabolism in *M. morganii*. This could include studies on the dose–response relationship of CFS antibacterial activity and the impact of different fermented food matrices on its inhibitory efficiency. This study provides insights into reduction of the accumulation of BAs and contributes to the safe production of fermented foods.

## Figures and Tables

**Figure 1 foods-12-03625-f001:**
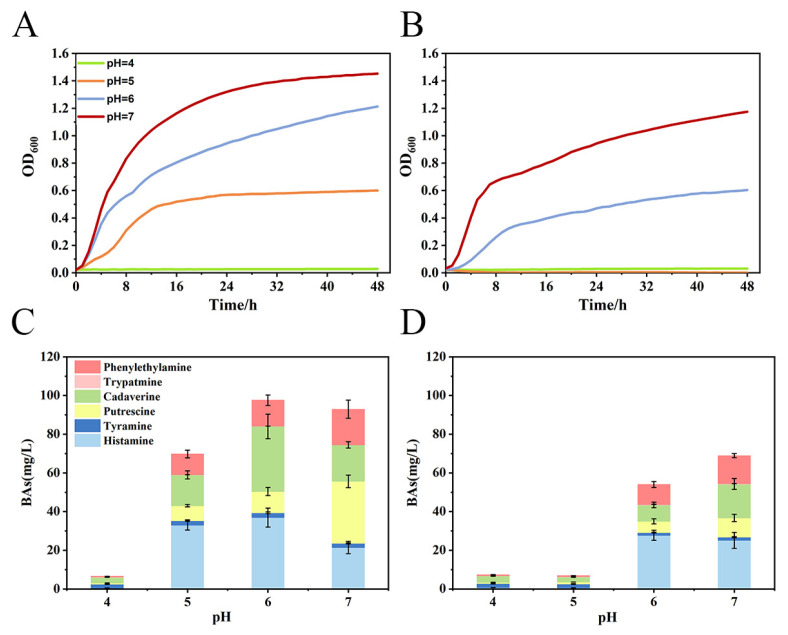
Effect of *Lactiplantibacillus plantarum* M1 cell-free supernatant (CFS) on the growth and biogenic amine production capacity of *Morganella morganii*. *M. morganii* cultured in blank medium was the control group, and *M. morganii* cultured in medium supplemented with *Lpb. plantarum* CFS was the experimental group; (**A**,**C**) the growth curves and biogenic amine production analyses of *M. morganii* under different pH conditions; (**B**,**D**) the effects of *Lpb. plantarum* CFS on the growth of *M. morganii* and the biogenic amine production under different pH conditions.

**Figure 2 foods-12-03625-f002:**
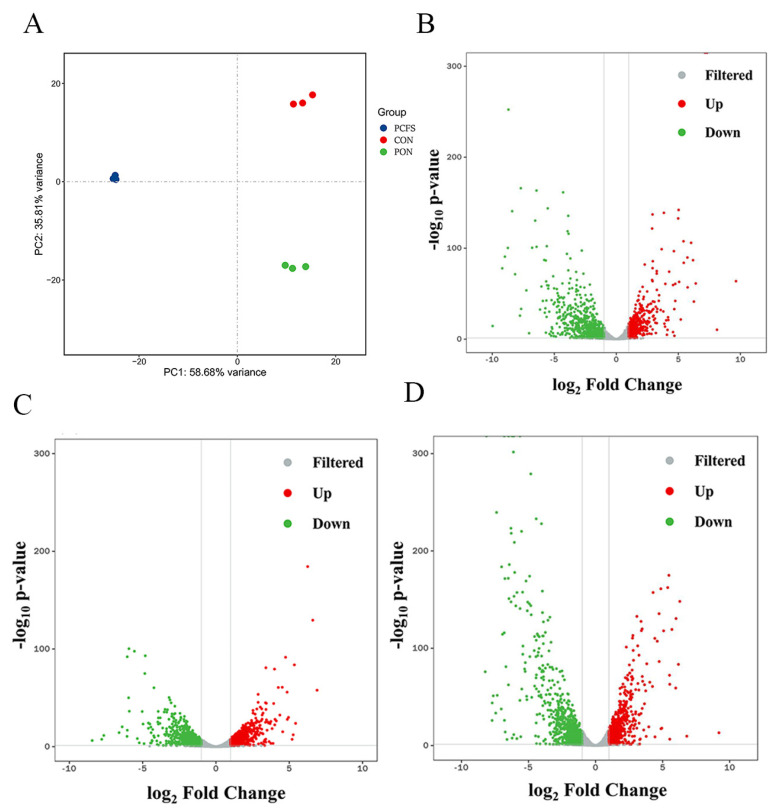
Transcriptome data and total analysis of differentially expressed genes (DEGs) in *M. morganii* treated with different conditions. Comparison of *M. morganii* under acidic environment culture (PON), *Lpb. plantarum* CFS and acidic environment culture (PCFS), and blanket culture (CON); (**A**): principal component analysis (PCA) plot; volcano plot between two: (**B**) PCFS vs. CON; (**C**) PON vs. CON; (**D**) PCFS vs. PON. Red: up−regulated genes, green: down−regulated genes, grey: non−significant genes.

**Figure 3 foods-12-03625-f003:**
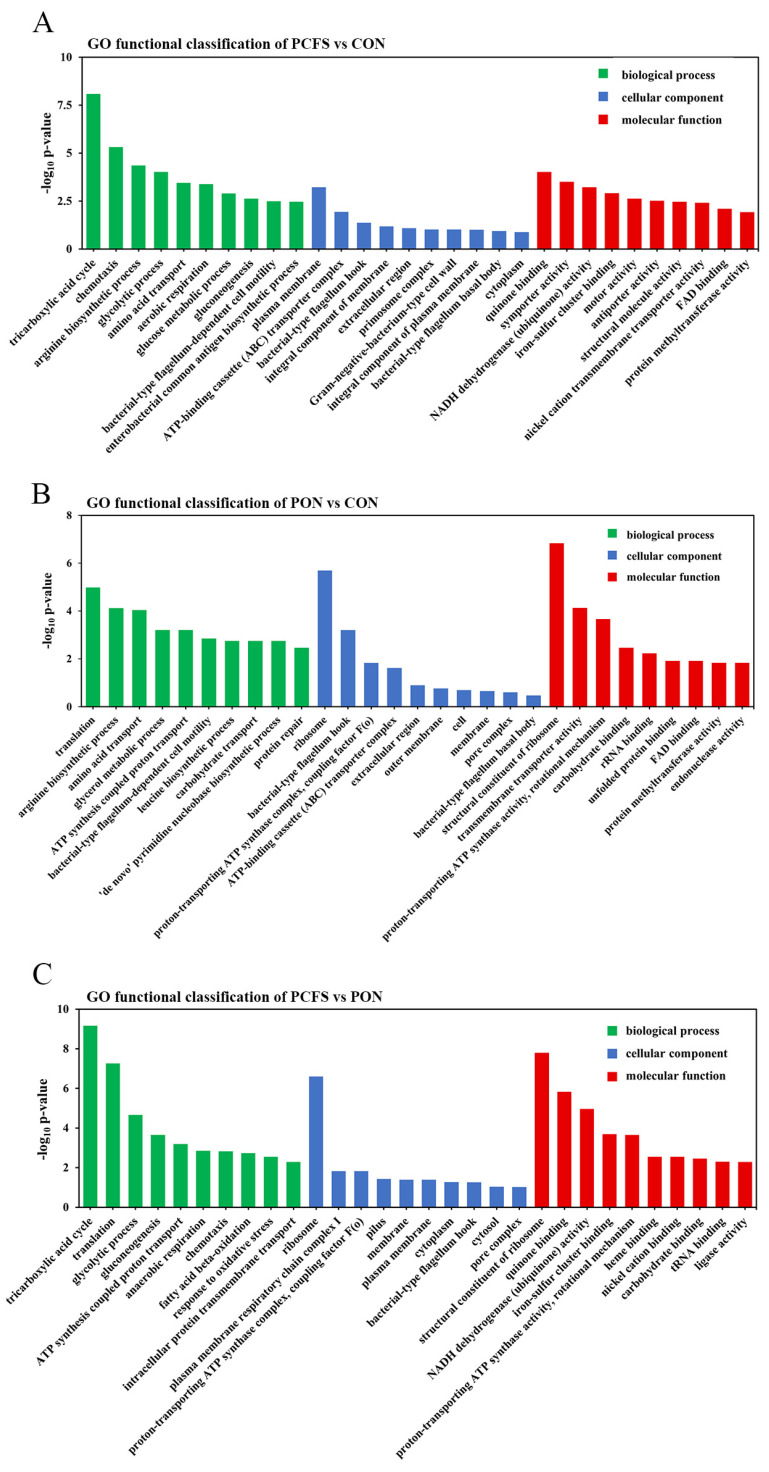
GO functional classification of DEGs. (**A**) PCFS vs. CON; (**B**) PON vs. CON; (**C**) PCFS vs. PON, with genes belonging to biological processes, cellular components and molecular functional categories indicated in green, blue, and red, and the top 10 GO terms with significant DEGs in each category are shown in the figure.

**Figure 4 foods-12-03625-f004:**
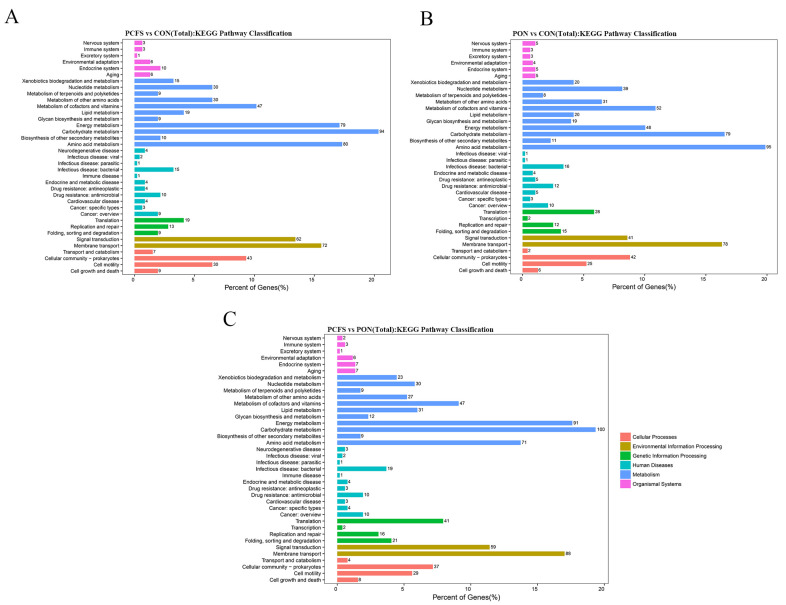
Classification of KEGG enrichment of DEGs. (**A**) PCFS vs. CON; (**B**) PON vs. CON; (**C**) PCFS vs. PON. Orange: cellular processes, yellow: environmental information processing, green: genetic information processing, cyan: human diseases, blue: metabolism, and pink: organic systems.

**Figure 5 foods-12-03625-f005:**
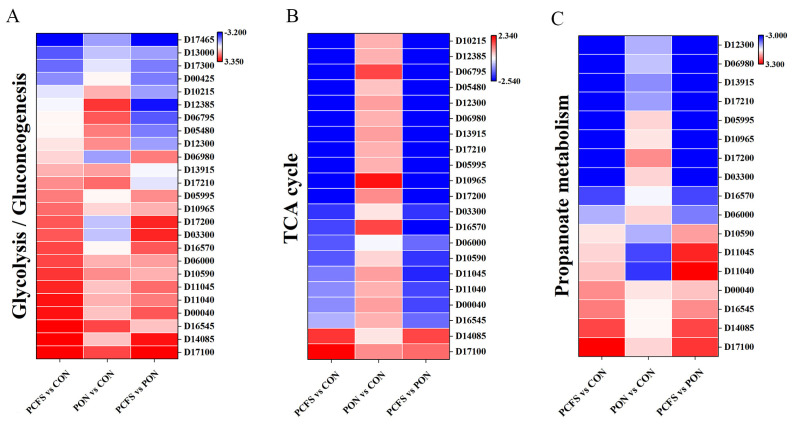
Heat map of typical DEGs associated with carbohydrate metabolism in different ways of culturing *M. morganii*. (**A**) Glycolysis; (**B**) tricarboxylic acid cycle; (**C**) propionate metabolism. Red: up−regulated genes, blue: down−regulated genes.

**Figure 6 foods-12-03625-f006:**
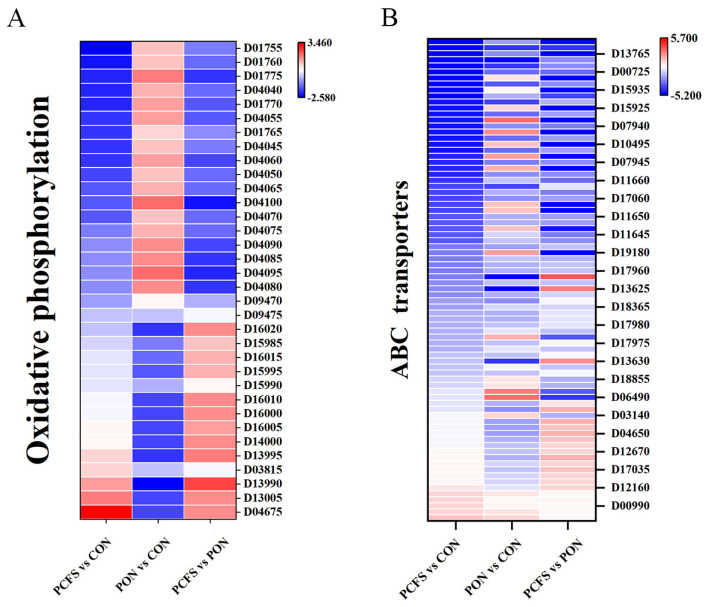
Effect of different treatments on energy metabolism and membrane transporters in *M. morganii*. (**A**) Oxidative phosphorylation; (**B**) ABC transporters. Red: up−regulated genes, blue: down−regulated genes.

**Figure 7 foods-12-03625-f007:**
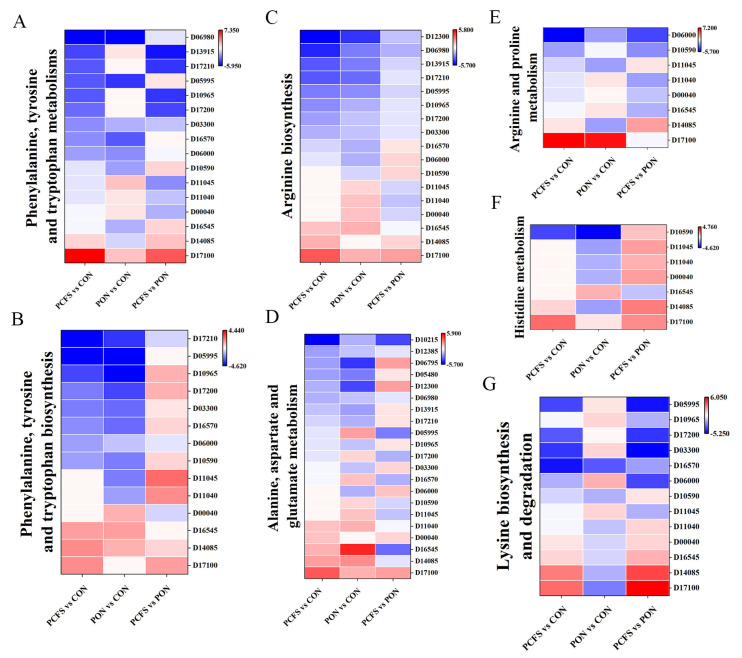
Heat map of typical DEGs associated with amino acid metabolism in different ways of culturing *M. morganii.* These include (**A**) phenylalanine, tyrosine, and tryptophan metabolism; (**B**) phenylalanine, tyrosine, and tryptophan biosynthesis; (**C**) arginine biosynthesis; (**D**) alanine, aspartate, and glutamate metabolism; (**E**) arginine and proline metabolism; (**F**) histidine metabolism; (**G**) lysine biosynthesis and degradation. Red: up−regulated genes and blue: down−regulated genes.

## Data Availability

The data that support the findings of this study are available from the corresponding author upon reasonable request.

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
