# Peer review of "Inhibition Mechanism of Lactiplantibacillus plantarum on the Growth and Biogenic Amine Production in Morganella morganii"

_foods, 2023, doi:10.3390/foods12193625_

Round 1

Reviewer 1 Report

Dear editor and authors,

1- The main question in this manuscript: Why did the authors use crude bacterial extract (supernatant )? What are the antioxidants in this CFS? How is the cause of inhibition of vital amines attributed?

2-How did the authors discover that the CFS has biological activity against other bacterial species?

3-The modern nomenclature for lactic acid bacteria must be used in the current study and future studies. Authors must write the new name of the bacteria in the title and throughout the manuscript (Lactiplantibacillus plantarum).

4-The introduction to the manuscript requires adding some sources that support the manuscript, so I suggest the following studies

Wang, Y., Pei, H., Liu, Y., Huang, X., Deng, L., Lan, Q., ... & Yang, Y. (2021). Inhibitory mechanism of cell-free supernatants of Lactobacillus plantarum on Proteus mirabilis and influence of the expression of histamine synthesis-related genes. Food Control, 125, 107982.‏

Al-Sahlany, S. T., & Niamah, A. K. (2022). Bacterial viability, antioxidant stability, antimutagenicity and sensory properties of onion types fermentation by using probiotic starter during storage. Nutrition & Food Science, 52(6), 901-916.‏

Sun, L., Guo, W., Zhai, Y., Zhao, L., Liu, T., Yang, L., ... & Duan, Y. (2023). Screening and the ability of biogenic amine-degrading strains from traditional meat products in Inner Mongolia. LWT, 176, 114533.

5-Many working methods do not contain scientific references that can be consulted and verified, such as: Preparation of L. plantarum cell-free supernatant (CFS), Effect of different conditions on the growth and BAs production of M. morganii, ...............etc. 

6- What inhibitory substances does the bacterial extract contain so that it can be mixed in a 1:1 ratio with the growth medium?

7-line 88, The authors used a bacterial inoculum volume of 2% of the target bacteria. How many viable colonies are there in this volume? Size does not mean anything. What is important here is knowing the viable numbers at this size because they will be subject to a process of inhibition, and the number is the influence.

8-line 103, Why are bacteria incubated for only 10 hours? Known in previous studies is 16-18 hours. 

9-line 113, The unit of count for bacteria is CFU, not a cell.

10-Figure 1a shows that the authors used four pH numbers, but the number of lines in the figure is 3.

Figure 1b What do the blue and red lines mean?

11-The conclusions must be rewritten because they contain many results and are not a conclusion from the study. 

Quality of English Language is good.

Reviewer 2 Report

The aim of this article is to evaluate the inhibitory effect of L. plantarum on the development and ability to produce biogenic amines of Morganella morganii, using classical methods but also using transcriptomic methodology to evaluate its possible mechanism. The approach is excellent, and the paper is well-written and organized. However, I would like to make a few observations.

The methodology does not include the characteristics and/or brand, place of manufacture, etc. of the equipment and reagents used.

Changing the conditions from rpm to g would be preferable, especially if the centrifuge used is unknown. The text contains many acronyms, so it should be checked if they have been previously defined in all cases. The y-axis should be scaled the same in Figures 1 and B. In Figure 1, C and I congratulate the authors for their graphic presentation, but it is difficult to see if there are significant differences in the production of biogenic amines. The quantitative data and the statistical study could be presented as a table as supplementary material.

Round 2

Reviewer 1 Report

Dear Editors, 

Authors did all necessary changes to improve the manuscript and now I recommend it for publication in the current form.